# The Interactions between Two Fungal Endophytes *Epicoccum layuense* R2-21 and *Alternaria alternata* XHYN2 and Grapevines (*Vitis vinifera*) with De Novo Established Symbionts under Aseptic Conditions

**DOI:** 10.3390/jof9121154

**Published:** 2023-11-30

**Authors:** Xiao-Xia Pan, Hui-Zhi Liu, Yu Li, Ping Zhou, Yun Wen, Chun-Xi Lu, You-Yong Zhu, Ming-Zhi Yang

**Affiliations:** 1School of Ecology and Environmental Science, Yunnan University, Kunming 650504, China; pan901805@126.com (X.-X.P.);; 2Key Laboratory of Chemistry in Ethnic Medicinal Resources, State Ethnic Affairs Commission & Ministry of Education, School of Ethnic Medicine, Yunnan Minzu University, Kunming 650504, China; 3Key Laboratory for Agro-Biodiversity and Pest Control of Ministry of Education, Yunnan Agricultural University, Kunming 650201, China

**Keywords:** fungal endophytes, *Epicoccum layuense* R2-21 (Epi R2-21), *Alternaria alternata* XHYN2 (Alt XHYN2), grapevine, transcriptomic analysis, endophyte–plant interaction

## Abstract

In this study, we focused on grapevine–endophyte interactions and reprogrammed secondary metabolism in the host plant due to defense against the colonization of endophytes. Thus, the transcriptional responses of tissue cultured grapevine seedlings (*Vitis vinifera* L. cv.: Cabernet Sauvignon) to two fungal endophytes *Epicoccum layuense* R2-21 (Epi R2-21) and *Alternaria alternata* XHYN2 (Alt XHYN2) at three different time points (6 h, 6 d, 15 d) were analyzed. As expected, a total of 5748 and 5817 differentially expressed genes (DEGs) were separately initiated in Epi R2-21 and Alt XHYN2 symbiotic tissue cultured seedlings compared to no endophyte treatment. The up-regulated DEGs at all time points in Epi R2-21- or Alt XHYN2–treated seedlings were mainly enriched in the flavonoid biosynthesis, phenylpropanoid biosynthesis, phenylalanine metabolism, stilbenoid, diarylheptanoid and gingerol biosynthesis, and circadian rhythm–plant pathways. In addition, the up-regulated DEGs at all sampling times in Alt XHYN2-treated tissue cultured seedlings were enriched in the plant–pathogen interaction pathway, but appeared in Epi R2-21 symbiotic seedlings only after 15 d of treatment. The down-regulated DEGs were not enriched in any KEGG pathways after 6 h inoculation for Epi R2-21 and Alt XHYN2 treatments, but were enriched mainly in photosynthesis–antenna proteins and plant hormone signal transduction pathways at other sampling times. At three different time points, a total of 51 DEGs (all up-regulated, 1.33–10.41-fold) were involved in secondary metabolism, and 22 DEGs (all up-regulated, 1.01–8.40-fold) were involved in defense responses in endophytic fungi symbiotic tissue cultured seedlings. The protein–protein interaction (PPI) network demonstrated that genes encoding CHS (*VIT_10s0042g00920*, *VIT_14s0068g00920*, and *VIT_16s0100g00910*) and the *VIT_11s0065g00350* gene encoding CYP73A mediated the defense responses, and might induce more defense-associated metabolites. These results illustrated the activation of stress–associated secondary metabolism in the host grapevine during the establishment of fungi–plant endophytism. This work provides avenues for reshaping the qualities and characteristics of wine grapes utilizing specific endophytes and better understanding plant–microbe interactions.

## 1. Introduction

Endophytes systematically colonize all plants in the world and establish a symbiotic or parasitic relationship with the host plant [1]. In endophyte–plant symbionts, the plants provide the habitat supporting the living of endophytes; in turn, endophytes influence the host plants physiologically and metabolically to enhance their environmental adaptation and growth [2,3].

Grapevines are one of the most cultivated fruit plants and harbor diverse endophytes that define the qualities and characteristics of grapes and wines [4]. Endophytes function as beneficial microorganisms or pathogens throughout the whole vine’s lifespan. Therefore, investigating the interactions between endophytes and grapevines is very meaningful, and beneficial endophytes are expected to be utilized in viticulture. However, the mechanisms of secondary metabolites induced by grapevine–endophyte interactions have few reports to date. Our previous studies have examined the biochemical effects of fungal endophytes on grapes and showed that endophytic fungi in grapes may contribute to anthocyanin production [5,6,7]. The total stilbene content of the grape cells increased when co-cultivated with endophytes [8]. Additionally, endophytic fungi isolated from *V. vinifera* possessed the capability to produce resveratrol [9,10]. These results raised the possibility of applying endophytes to reshape the metabolite profile of grapevine. In addition, the intricate environmental factors reshape the microbiota in the inner grapevine [11]. It is difficult to exactly determine the detailed interactions between certain endophytes and grapevines in vivo. Symbionts of grapevines with known endophyte colonization are good materials for this purpose.

In recent years, transcriptomic analyses have been widely applied to reveal changes in the expression levels of various genes in pathogen-attacked grapevines. Haile et al. demonstrated the cross-talk between *Vitis vinifera* cv. Pinot Noir berry and pathogen *Botrytis cinerea* by transcriptome and metabolic analysis, and berries responded by differentially regulating genes involved in the monolignol, flavonoid, and stilbenoid biosynthesis pathways [12]. Multiomics analyses were conducted to explore the interaction between grape berries of ‘Carignan’ and powdery mildew, and integrated genes involved in biotic stress responses, and pathways were modulated in response to powdery mildew infection [13]. Another study revealed the molecular mechanisms underlying the interaction between grapevine and the pathogen *Plasmopara viticola*, identifying the susceptibility genes and pathways involved in plant–pathogen interactions [14]. However, transcriptome analysis of the fungal endophyte-mediated shaping of the grapevine metabolite pathway has not been performed.

Therefore, this paper analyses the different responses of tissue seedlings from the variety ‘Cabernet Sauvignon’ (*Vitis vinifera* L. cv., the worldwide planted wine grape variety) with two fungal endophytes *Epicoccum layuense* R2-21 (Epi R2-21) and *Alternaria alternata* XHYN2 (Alt XHYN2) inoculation at three different time points (6 h, 6 d, 15 d) and compared the molecular responses at transcriptomic levels in tissue cultured grapevine seedlings after co-cultivated with the fungi. The data suggests the activation of defensive mechanisms in response to fungal endophytes Epi R2-21 and Alt XHYN2 inoculation, and thus might intrigue more defense-associated metabolites in tissue cultured grapevine seedlings (*Vitis vinifera* L. cv.: Cabernet Sauvignon).

## 2. Materials and Methods

### 2.1. Preparation of In Vitro Cultured Grapevine Seedlings and Fungal Endophytes

The tender stem-segments with auxiliary buds as the explant were applied to establish in vitro cultured grapevine plantlets (single bud clones, *Vitis vinifera* L. Cabernet Sauvignon) as previously described [15], with the modified disinfection procedure: 75% C_2_H_5_OH for 30 s, and 10% NaClO for 20 min. Murashige and Skoog (MS) medium [16] was applied for tissue culture plantlets cultivation, containing 3% sucrose (*m*/*v*), 0.75% agar, and supplemented with vitamins (myo-inositol 100 mg L^−1^, pyridoxine HCl 1 mg L^−1^, thiamine HCl 1 mg L^−1^, nicotinic acid 1 mg L^−1^, D-calcium pantothenate 1 mg L^−1^ and biotin 0.01 mg L^−1^). Tissue culture plantlets were cultured in a light culture room at 26 °C with 12/12 light/dark cycles for 2 months, with 7 to 10 expanded leaves prepared for the following experiments.

The endophytic fungi Epi R2-21 and Alt XHYN2 were isolated from grapevine leaves from local vineyards (Kunming, Yunnan Province, China) (Appendix A). The isolation of fungal endophytes followed the tissue patch method [17], and purified fungal strains were identified using internal transcribed spacer (ITS) DNA sequences [18]. Total DNA of fungal isolates was first extracted with cetyltrimethylammonium bromide (CTAB) methods, and then ITS sequences were amplified with primer pairs ITS4 and ITS5 [19].

### 2.2. Inoculation of Endophytic Fungi and Determination of the Isolation Rates of Fungal Endophytes

Fungal strains were cultured on glass paper, which was covered on potato dextrose agar medium (PDA: peeled and diced potato 200 g, dextrose 20 g, agar 15 g and water 1 L) for 3–5 d, and fungal growth masses were harvested and suspended in 0.9% normal saline (final concentration was 2.5 g/L). Tissue seedlings were smear inoculated with a sterilized cotton swab with suspensions of fungal growth masses of Epi R2-21 and Alt XHYN2 to establish endophyte–host symbionts. Tissue cultured seedlings without endophyte inoculation were used as controls. The controls and treatments were cultured at 26 °C for 6 h, 6 d, and 15 d, respectively. All samples of seedlings were harvested, and leaf surface fungi were washed off with sterile water, immediately frozen in liquid nitrogen, and stored in −80 °C freezers for further assays. The leaves were harvested after 15 d inoculation to detect the isolation rates of fungal endophytes using the tissue patch method [17], and fungal colonies were identified using ITS DNA sequences. These samples were defined as R2-21_6h, R2-21_6d, and R2-21_15d (leaf samples from Epi R2-21 treatments were collected at 6 h, 6 d and 15 d after inoculation); XHYN2_6h, XHYN2_6d, and XHYN2_15d (leaf samples from Alt XHYN2 treatments were collected at 6 h, 6 d and 15 d after inoculation); and Con_6h, Con_6d, and Con_15d (leaf samples from the control were collected at 6 h, 6 d and 15 d after inoculation).

### 2.3. RNA Sequencing, Library Construction, and Quantitative Real-Time PCR Validation (qRT–PCR)

Total RNA was extracted using TRIzol^®^ Reagent according to the instructions (Invitrogen, Carlsbad, CA, USA) and genomic DNA was removed using DNase I (Takara, Tokyo, Japan). The quantity was evaluated by 1% agarose gel electrophoresis and a 2100 Bioanalyzer (Agilent technologies, Santa Clara, CA, USA). The RNA samples with qualified purity were obtained using a NanoDrop 2000 spectrophotometer (Thermo, Waltham, MA, USA). The RNA-seq library was constructed following the TruSeq^TM^ RNA sample preparation kit from Illumina (San Diego, CA, USA). The paired-end RNA-seq sequencing library was sequenced with the Illumina HiSeqxten/NovaSeq 6000 sequencer (2 × 150 bp read length). Raw data were cleaned by trimming the adapter sequences, and the clean reads were separately aligned against the reference grapevine genome with the orientation mode using HISAT2 version 2.1.0. The quality of the RNA-Seq data was validated by performing qRT–PCR analysis on an ABI7500 fluorescent quantitative PCR machine (Applied Biosystems, Waltham, MA, USA). Total RNA was extracted and purified from leaves of tissue grapevine seedlings as above, and three biological replicates were performed on each sample. The results were normalized to the reference gene *EF1*, and the relative gene expression was calculated based on the 2^−ΔΔCt^ method. All primers are listed in Appendix A. Three biological replicates were conducted with at least 10 plants for each replicate.

### 2.4. Identification of Differentially Expressed Genes (DEGs), Functional Annotation, and Enrichment Pathway Analysis

The fragments per kilobase of transcript per million mapped reads (FPKM) method was used to investigate differential gene expression at different time points, and the FPKM of each gene was calculated based on the length of the gene and read counts mapped to the gene. DESeq2 in Bioconductor was utilized for DEGs analysis. Significant DEGs were identified using adjusted *p*-adjust < 0.05 and |log_2_FC| ≥ 1 as the cut-off criteria. The Benjamini–Hochberg test was applied to obtain the significance of differential gene expression. Gene Ontology (GO) functional enrichment and Kyoto Encyclopedia of Genes and Genomes (KEGG) pathway analysis were carried out by Goatools.

### 2.5. Constructing the Protein–Protein Interaction (PPI) Network to Screen Key Genes

The PPI network was constructed based on the data produced by the Search Tool for the Retrieval of Interacting Genes database (http://string-db.org/ (accessed on 25 October 2023)) and visualized using Cytoscape software (version 3.4.1). The confidence score cut-off applied for interactions was 0.4. Node color and size were determined based on betweenness centrality and degree, respectively.

## 3. Results

### 3.1. Fungal Endophytes Epi R2-21 and Alt XHYN2 Could Lead to Infection and Symbiosis in Tissue Cultured Grapevine Seedlings

After 15 d of co-culture with Epi R2-21 and Alt XHYN2, the grapevine seedlings maintained a good physiological state (Appendix A), and Epi R2-21 and Alt XHYN2 inoculation led to successful symbiosis in tissue cultured seedlings. The isolation rates of fungal endophytes in grapevine seedlings were 28% and 34%, respectively. However, the MS media inoculated with Epi R2-21 or Alt XHYN2 were slightly off-colored, with a little brown (Appendix A). There may be two reasons for this: firstly, the suspensions of Epi R2-21 and Alt XHYN2 may have fallen onto the medium, causing the growth of fungal mycelia on the plate; secondly, inoculation with Epi R2-21 or Alt XHYN2 may have caused the secretion of some colored secondary metabolites.

### 3.2. Transcriptomic Analysis of Tissue Cultured Seedlings in Response to Endophytes at Different Time Points

Transcriptomic changes in response to endophytes were examined in leaves of tissue cultured seedlings after 6 h, 6 d, and 15 d inoculation with Epi R2-21 and Alt XHYN2. High-quality clean reads ranged from 50.8 to 66.1 million for 27 samples. The GC percentages for these clean reads varied from 45.96% to 46.70%, and the Q20 and Q30 percentages of all libraries were >95% (Appendix A), which indicated the high quality of RNA and sequence. Based on the alignment with the grapevine reference genome, the mapping ratios for the controls and the treatments were 90.51% to 93.41%. A total of 33,481 genes were identified, comprising 29,971 known and 3510 novel genes (Appendix A). A total of 17,302 to 18,063 genes were identified, and 15,500 genes were common to all of the samples (Appendix A). The correlation coefficients of the relationships of all the transcriptomes ranged from 0.88 to 0.98 (Appendix A). RNA-seq data were validated by qRT–PCR. The similar expression trends of the RNA-Seq data, except in genes *VIT_16s0039g01300*, *VIT_11s0065g00350*, *VIT_10s0003g00470*, and *VIT_05s0094g00200* in sample XHYN2_15d, indicated the reliability of the transcriptomic data (Appendix A).

Principal component analysis (PCA) revealed that the 27 samples clustered in 9 corresponding discrete groups, and each replicate from the same group was clustered closely together (Figure 1A). The sampling groups Con_6h, R2-21_6h and XHYN2_6h were closely clustered. In contrast, Con_6d and Con_15d, R2-21_6d and XHYN2_6d, and R2-21_15d and XHYN2_15d showed two and two groups clustered together, which indicated that endophyte inoculation induced significant gene changes in tissue cultured seedlings.

A total of 5748 DEGs were identified in tissue cultured seedlings after 6 h, 6 d, and 15 d of inoculation with Epi R2-21. Briefly, 1810 (1346 up- and 464 down-regulated), 3316 (1945 up- and 1371 down-regulated), and 4142 (2784 up- and 1358 down-regulated) DEGs were identified after 6 h, 6 d, and 15 d of inoculation with Epi R2-21, respectively (Figure 1B). A total of 5817 DEGs after 6 h, 6 d, and 15 d inoculation with Alt XHYN2. Briefly, 1937 (1708 up- and 229 down-regulated), 3284 (1991 up- and 1293 down-regulated), and 5506 (3120 up- and 2386 down-regulated) DEGs were identified after 6 h, 6 d, and 15 d of inoculation with Alt XHYN2, respectively (Figure 1B). The distribution of up- and down-regulated genes was calculated for each time point and is presented in a Venn diagram (Figure 1C,D). Although a unique set of genes increased at each time point (with a total of 1688), the expression levels of a large number of genes (442) were significantly up-regulated at three different time points. In addition, a unique set of genes was significantly down-regulated at each time point (total of 1418), and only 22 genes showed decreased expression at three different time points.

### 3.3. GO and KEGG Enrichment Analysis of DEGs

GO analysis was used for the functional classification of the DEGs in tissue cultured seedlings after 6 h, 6 d, and 15 d of inoculation with Epi R2-21 and Alt XHYN2. The top 20 enriched GO terms of all DEGs are shown in Appendix A. After inoculation at 6 h, 6 d, and 15 d with Epi R2-21 or Alt XHYN2, within the biological process category, DEGs were all mainly enriched in the metabolic process, cellular process, and single-organism process. Within the molecular function category, DEGs were all mainly enriched in catalytic activity and binding. Within the cellular category, DEGs activated by endophyte inoculation were cell, cell part, membrane, and membrane part.

Then, KEGG enrichment analysis of up- and down-regulated DEGs was performed (Figure 2 and Figure 3), and the pathways with the greatest up- and down-regulation of DEGs are listed in Table 1. After 6 h, 6 d, and 15 d inoculation with Epi R2-21, the up-regulated DEGs were significantly enriched in 12, 18, and 17 pathways, respectively (Figure 2A–C). The down-regulated DEGs were significantly enriched in 5 and 3 pathways after 6 d and 15 d of inoculation, respectively (Figure 3B,C). For Alt XHYN2, the up-regulated DEGs were significantly enriched in 10, 19, and 19 pathways, respectively (Figure 2D–F). The down-regulated DEGs were significantly enriched in 6 and 7 pathways after 6 d and 15 d of inoculation, respectively (Figure 3E,F). None of the down-regulated DEGs had significantly enriched pathways after 6 h of inoculation in both the Epi R2-21 or Alt XHYN2 treatments (*p*-adjust > 0.05) (Figure 3A,D).

For both the Epi R2-21 and Alt XHYN2 treatments at all time points, the largest number of up-regulated DEGs was enriched in flavonoid biosynthesis, phenylpropanoid biosynthesis, phenylalanine metabolism, and the stilbenoid, diarylheptanoid and gingerol biosynthesis, and circadian rhythm–plant pathways (Figure 2; Table 1). In addition, the majority of the up-regulated DEGs were enriched in plant–pathogen interactions in Epi R2-21 treatments after 15 d inoculation and in Alt XHYN2 treatments at all time points, enriched in plant hormone signal transduction in Epi R2-21 treatments after 15 d inoculation and in Alt XHYN2 treatments after 6 d inoculation, and enriched in MAPK signaling pathway in Epi R2-21 treatments after 15 d inoculation, and in Alt XHYN2 treatments after 6 d and 15 d inoculation. The down-regulated DEGs were not enriched in the KEGG pathway after 6 h inoculation for either the Epi R2-21 or Alt XHYN2 treatments, while the down-regulated DEGs were mainly enriched in photosynthesis–antenna proteins and plant hormone signal transduction after 6 d and 15 d inoculation (Figure 3; Table 1).

### 3.4. Fungal Endophytes Epicoccum layuense R2-21 and Alt XHYN2 Induced the Expression of Genes Involved in Secondary Metabolism in Tissue Cultured Grapevine Seedlings

Secondary metabolism in tissue cultured seedlings was also extensively reprogrammed in response to Epi R2-21 and Alt XHYN2 at different time points (Figure 4, Appendix A). Plant secondary metabolic pathways play significant roles in plant defense by enforcing or producing physical and chemical barriers against pathogen infection. After 6 h, 6 d, and 15 d of inoculation with Epi R2-21 and Alt XHYN2, 106 DEGs (82 up- and 24 down-regulated) were involved in the phenylpropanoid biosynthesis pathway (Appendix A), 70 DEGs (58 up- and 12 down-regulated) were involved in the flavonoid biosynthesis pathway (Appendix A), 41 DEGs (37 up- and 4 down-regulated) were involved in the phenylalanine metabolism pathway (Appendix A), and 28 DEGs (26 up- and 2 down-regulated) were involved in the stilbenoid, diarylheptanoid, and gingerol biosynthesis pathway (Appendix A). Among the above DEGs, 51 genes were commonly and differentially expressed across the time points (all up-regulated, 1.33–10.41-fold), including 1 gene encoding trans-cinnamate 4-monooxygenase (CYP73A), 11 genes encoding phenylalanine ammonia-lyase (PAL), 22 genes encoding chalcone synthase (CHS), 3 genes encoding stilbene synthase (STS), 6 genes commonly encoding CHS and STS, 2 genes encoding peroxidase (POD), 1 gene encoding caffeic acid 3-*O*-methyltransferase (COMT), 1 gene encoding ferulate-5-hydroxylase (F5H, CYP84A), 1 gene encoding caffeoyl-CoA *O*-methyltransferase (CCoAOMT), and 1 gene encoding flavonoid 3′,5′-hydroxylase (F3′5′H) (Appendix A). Furthermore, the *VIT_11s0065g00350* gene encoding CYP73A was involved in flavonoid biosynthesis, phenylpropanoid biosynthesis, phenylalanine metabolism, and the stilbenoid, diarylheptanoid and gingerol biosynthesis pathways; 12 genes (*VIT_16s0039g01240*, *VIT_16s0039g01110*, *VIT_16s0039g01280*, *VIT_16s0039g01300*, *VIT_00s2849g00010*, *VIT_16s0039g01360*, *VIT_16s0039g01120*, *VIT_16s0039g01170*, *VIT_00s2508g00010*, *VIT_16s0039g01100*, *VIT_16s0039g01130*, *VIT_08s0040g01710*) encoding PAL were involved in phenylpropanoid biosynthesis and phenylalanine metabolism pathways; and 6 genes (*VIT_16s0100g00810*, *VIT_16s0100g00960*, *VIT_16s0100g00950*, *VIT_16s0100g00860*, *VIT_16s0100g01130*, *VIT_10s0042g00870*) commonly encoding CHS and STS were involved in flavonoid biosynthesis, stilbenoid, diarylheptanoid, and gingerol biosynthesis pathways.

### 3.5. Fungal Endophytes Epicoccum layuense R2-21 and Alt XHYN2 Induced the Expression of Genes Involved in Defence Responses in Tissue Cultured Grapevine Seedlings

Notably, a large number of DEGs involved in the pathways related to defense responses were induced in tissue cultured seedlings after 6 h, 6 d, and 15 d of inoculation with Epi R2-21 and Alt XHYN2, such as plant–pathogen interaction, plant hormone signal transduction, plant MAPK signaling pathway, and brassinoid steroid biosynthesis pathways. In particular, the plant–pathogen interaction and the plant hormone signal transduction pathways induced the most DEGs (Figure 5 and Figure 6).

We identified 120 DEGs (84 up- and 36 down-regulated) in tissue cultured seedlings after 6 h, 6 d, and 15 d inoculation with Epi R2-21 and Alt XHYN2 that were involved in the plant–pathogen interaction pathway (Appendix A); 119 DEGs (62 up- and 57 down-regulated) were involved in the plant hormone signal transduction pathway (Appendix A). Among the above DEGs, 22 genes were commonly differentially expressed across the time points (all up-regulated, 1.01–8.40-fold) (Appendix A). In the plant–pathogen interaction pathway, 11 genes encode calmodulin-like protein CML, 1 gene encodes disease resistance protein RPM1, 1 gene encodes transcription factor WRKY33, 2 genes encode transcription factor WRKY22, 1 gene encodes calcium–dependent protein kinase CPK29, and 1 gene encodes respiratory burst oxidase RBOH. In the plant hormone signal transduction pathway, two genes encode the auxin-responsive GH3 gene family GH3, one gene encodes the SAUR family protein SAUR, and one gene encodes the transcription factor TGA.

### 3.6. Regulatory Network Analysis of Key Genes Involved in Secondary Metabolism against Endophyte Defence in Tissue Cultured Grapevine Seedlings

To explore the possible secondary metabolic pathway in grapevine defense against endophyte and select key genes involved in producing stress-associated secondary metabolites, the above 73 genes that were involved in secondary metabolism and defense responses and commonly differentially expressed across the time points, were used to build a PPI network with *Vitis vinifera*. There are 2 clusters identified in our constructed network containing 38 nodes, each representing 1 protein and connected by 300 edges (Figure 7). In cluster I, the genes *VIT_10s0042g00920*, *VIT_14s0068g00920*, and *VIT_16s0100g00910*, which encode CHS involved in flavonoid biosynthesis, had the highest scores for betweenness centrality, indicating that they are most important in response to endophytes defense to produce secondary metabolites. In addition, in cluster II, the *VIT_11s0065g00350* gene is connected with cluster I by the *VIT_10s0042g00920* and *VIT_16s0100g01040* genes. The *VIT_11s0065g00350* gene, encoding CYP73A, was commonly involved in flavonoid biosynthesis, phenylpropanoid biosynthesis, phenylalanine metabolism, and the stilbenoid, diarylheptanoid, and gingerol biosynthesis pathways. Thus, the above genes were suggested to play crucial roles in the defense of endophytes and produce metabolites.

## 4. Discussion

Endophytes and plants gradually establish mutualistic and symbiotic relationships during interactions. According to the plant–endophyte coevolution hypothesis [20], the host plant can synthesize stress-induced secondary metabolites to defend themselves against pathogens with endophyte assistance. Endophytic fungi can promote the accumulation of bioactive secondary metabolites in medicinal plants of *Salvia abrotanoides,* with a positive effect on cryptotanshinone production [21]. Endophytic fungi isolated from grapevines demonstrated resveratrol-producing ability [22]. In our study, the molecular responses at transcriptomic levels in tissue cultured seedlings after 6 h, 6 d, and 15 d inoculation with Epi R2-21 and Alt XHYN2, which could be further explored to understand the mechanisms of the interaction relationship between grapevine and endophytic fungi, and further implications of the use of fungal endophytes in grape quality and character management.

### 4.1. Grapevine Transcriptomes Were Rapidly and Significantly Regulated in Tissue Cultured Grapevine Seedlings in Response to Fungal Endophytes Epi R2-21 and Alt XHYN2 Inoculation

Biotic and abiotic stress activate rapid responses in plants. For example, at 4 weeks post-inoculation, 599 genes of grapevine were differentially expressed due to *Botrytis cinerea* infection, whereas this number increased to 2296 at 3 months post-inoculation [12]. The most DEGs were recorded in the grapevine leaves after 1 d of inoculation with downy mildew, *Plasmopara viticola*, and a total of 6393 DEGs were obtained in the *Vitis* interspecific hybrid variety Bianca [23]. A total of 5251 genes were induced by the pathogen *Plasmopara viticola* after 3 d of inoculation in 3 cultivars of grapevine leaves [13]. Similarly, rapid and strong responses were induced by Epi R2-21 and Alt XHYN2 in grapevine leaves in our study. After 6 h, 6 d, and 15 d of inoculation, the transcript levels of 1810, 3316, and 4142 DEGs were significantly induced respectively by Epi R2-21, while 1937, 3284, and 5506 DEGs were regulated respectively by Alt XHYN2. The induced DEGs together with the obtained results in the plant–pathogen interaction pathway, suggested a stronger response triggered by Alt XHYN2. This was most likely due to Alt XHYN2 belonging to the fungal pathogen *Alternaria*, which lives in a mutual relationship or as a pathogen causing disease in the host plant [18].

### 4.2. Endophyte Symbiosis Induces Defence Responses in Tissue Cultured Grapevine Seedlings

In the plant–pathogen interaction pathway, as an important family of Ca^2+^ signaling systems, calmodulin (CALM), calmodulin-like proteins (CML), and calcium-dependent protein kinase (CDPK) produce reactive oxygen species (ROS) and nitric oxide (NO) separately to induce plant defense responses, including growth and development, biotic and abiotic stress, and plant hormone signal transduction [24]. Research has shown that cytoplasmic Ca^2+^ rapidly accumulates when plants perceive pathogen-associated molecular patterns (PAMPs), and Ca^2+^ is sensed by CALM as an important second messenger, forms a complex, binds to transcription factors, and then regulates downstream target genes [25].

In grapevine, Ca^2+^ influx was one of the defense reactions when inoculated with *Botrytis cinerea* [26]. Zhang et al. showed that cyclic nucleotide-gated channels (CNGCs), *CDPK*, and *RBOH* genes were up-regulated in grapevine infected with *Lasiodiplodia theobromae* [27]. Our data showed that the *CALM*, *CDPK*, and *CML* genes as early response genes for resistance, and the transcription factors WRKY22, WRKY33, and heat shock protein 90 (HSP90) were significantly up-regulated in the tissue cultured seedlings after 6 h, 6 d, and 15 d of inoculation with Epi R2-21 and Alt XHYN2. Some disease resistance proteins, such as RPM1, RPS2, and FLS2, were down-regulated during the later time points. This result indicated that the resistance response in the tissue cultured seedlings was stronger in the early stage of infection, while with the balance of antagonism between plants and endophytes, the defense responses decreased after the colonization of endophytes in the tissue cultured seedlings.

Additionally, the plant hormone signal transduction pathway is extremely vital for plant growth and adaptation to stresses [27]. In our study, plant hormone signal transduction and plant–pathogen interaction pathways were identified with 119 and 120 DEGs, respectively. *Aux/IAA* genes, *SAUR* genes, and *GH3* genes, as a group of primary auxin-response genes, participate in the regulation of plant growth, development, and adversity stress adaptation by regulating the expression of some genes [28]. The SAUR family, with the capacity to fine-tune growth in response to internal and external signals, has been shown to bind to calcium/calmodulin, suggesting a role for calcium ions in auxin signaling [29,30]. Several other *GH3* genes encode enzymes that conjugate amino acids to IAA and SA, suggesting a role for *GH3* genes in IAA–SA interactions [31,32]. Some studies have demonstrated that genes involved in hormone signal transduction, such as *Aux/IAA*, *GH3*, *SAUR*, *JAZ*, and *ARF*, are significantly up-regulated in grapevine by the fungal pathogen *Lasiodiplodia theobromae* [27]. Similarly, we found that Epi R2-21 and Alt XHYN2 inoculation significantly increased the expression levels of several genes related to signaling pathways, such as *GH3*, *SAUR*, *JAZ*, *TCH4*, and *BSK*. However, genes encoding IAA, ARF, AHP, CYCD3, SNRK2, PP2C, TIR1, and CRE1 had variable expression levels, with the majority being up-regulated during infection, but some also being down-regulated during the later time points, which suggests that these genes may be fine-tuned during defense responses, as Epi R2-21 or Alt XHYN2 form different and complex patterns of symbiosis within the host grapevine, and the growth of host plants is somewhat inhibited during the infection of endophytes. These outcomes were consistent with the DEGs involved in the plant–pathogen interaction pathway. The optimal defense theory predicts that the secondary metabolites within plants function as defense, and the production of these defense compounds entails growth costs. It seems to be advantageous for plants not to produce defense compounds beforehand but rather to produce them once they suffer biotic and abiotic stress [33].

### 4.3. Fungal Endophyte Inoculation Resulted in the Secondary Metabolite Profile Modification in Tissue Cultured Grapevine Seedlings

Secondary metabolites are all found to serve defensive functions (against microbes, viruses, herbivores, or other competing plants) or act as signaling molecules [34,35]. Generally, grapevines have adaptive mechanisms to combat abiotic and biotic stress (fungi, bacteria, viruses, or insect attack), which attests to the ability of the host plant to produce the same or similar bioactive metabolites.

Through RNA-seq enrichment pathway analysis, we found various pathways that focus on metabolic profiles involved in plant defense differently across time points after 6 h, 6 d, and 15 d of inoculation with Epi R2-21 and Alt XHYN2. For example, the phenylpropanoid biosynthesis, flavonoid biosynthesis, phenylalanine metabolism, stilbenoid, diarylheptanoid and gingerol biosynthesis, and sesquiterpenoid and triterpenoid biosynthesis pathways were all significantly enriched in tissue grapevine seedlings, suggesting that the regulatory genes from these pathways were rapidly and coordinately triggered in response to fungal endophyte infection.

The phenylalanine metabolic pathway including the phenylpropane metabolic pathway and isoflavone synthesis metabolic pathway, is one of the most important pathways of plant secondary metabolism. PAL, CHS, and CHI, as important enzymes in the pathway, catalyze the synthesis of flavonoids and other secondary metabolites [36]. STS is the key enzyme leading to the biosynthesis of resveratrol and stilbenes. Studies conducted on tissue cultured seedlings have shown that stilbenes could strongly accumulate in response to stresses [37]. In this study, we demonstrated that these genes encoding PAL, CHS, CHI, and STS were greatly activated in tissue cultured seedlings by Epi R2-21 or Alt XHYN2.

The phenylpropanoid biosynthesis pathway provides anthocyanins, flavonoids, various flavonoid, and isoflavonoid inducers, etc. [38,39]. In our study, a large number of DEGs were enriched and up-regulated in the phenylpropanoid biosynthesis pathway, including the genes encoding the enzyme PAL and enzymes involved in lignin biosynthesis, such as POD, bGLS-A, CAD, 4CL, and COMT, all of which are involved in inducing physical and chemical barriers against microbe infection, as well as secreting signal molecules that can ultimately induce defense-related gene expression in plants [27,40]. These outcomes were consistent with a previous study in which the PAL and enzymes involved in lignin biosynthesis showed increased expression when treated with the fungal pathogen *Lasiodiplodia theobromae* [27], which indicated that grapevine may employ the phenylpropanoid biosynthesis pathway to form lignin or other small signal molecules to limit Epi R2-21 and Alt XHYN2 invasion.

Flavonoids are a large and diverse group of secondary metabolites synthesized through a specific branch of the phenylpropanoid pathway [41,42,43]. As the key enzymes in the flavonoid biosynthesis pathway, CHS, F3′5′H, CYP73A, STS, F3H, FLS, and CCoAOMT play important roles in the growth and development of plants, and are related to the formation of plant flower color, UV damage prevention, disease resistance, pollen fertility, nodule formation, etc. [44]. In our study, DEGs encoding the above enzymes were activated by Epi R2-21 or Alt XHYN2. Similar results were also reported in the grapevine leaves inoculated with downy mildew, *Plasmopara viticola*, in which the expression of the genes encoding STS and CHS were strongly induced [37,45,46,47]. After being treated with a fungal elicitor, suspension-cultured bean cells rapidly increased the rate of synthesis of 3 enzymes of phenylpropanoid biosynthesis, PAL, CHS, and CHI, the mRNA amount reached the highest at 4 h, and the content increased by approximately 100 times [48]. As proposed by the optimal defense theory, secondary metabolites function as defense and the production of secondary metabolites reduces the plant growth as an ‘opportunity cost’. When plants produce secondary metabolites, the defense-obtained secondary metabolites are produced when behavioral gains outweigh growth gains. Plant growth slows down under environmental stress conditions, the cost of producing defensive secondary metabolites is lower, and plants synthesize more secondary metabolites [40].

## 5. Conclusions

In the present work, transcriptome analysis was performed to compare the status of endophytic and non-endophytic fungi in tissue grapevine seedlings at three time points to elaborate the metabolite responses after artificial inoculation with the fungal endophytes Epi R2-21 or Alt XHYN2. After 6 h, 6 d, and 15 d inoculation, transcriptome evaluation showed that grapevine genes are rapidly and significantly regulated in response to Epi R2-21 or Alt XHYN2 inoculation compared to the control, resulting in secondary metabolite profile modification. Taken together, the results reported here suggest that the induction of secondary metabolite biosynthesis may contribute to defense responses and therefore could reach a balance between grapevine and its symbiotic fungi. Our data reveal the mechanisms of the interactive relationship between grapevine and endophytic fungi, which provides new insight into the regulatory elements that might reshape the qualities and characteristics of wine grapes using endophytes.

## Figures and Tables

**Figure 1 jof-09-01154-f001:**
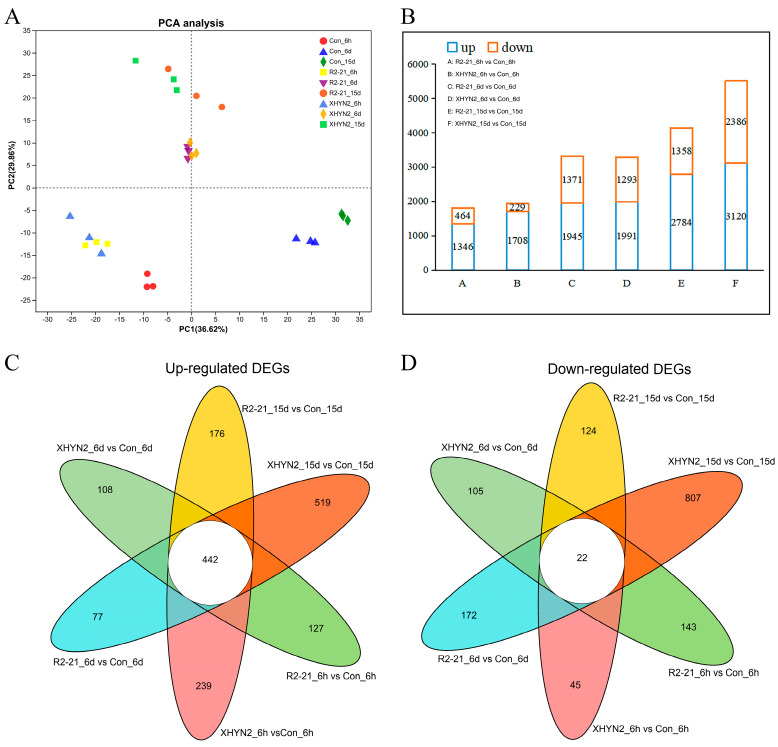
PCA (**A**), DEGs analysis (**B**), and Venn analysis (**C**,**D**) of the tissue cultured seedlings after 6 h, 6 d, and 15 d inoculation with Epi R2-21 or Alt XHYN2. DEGs were analyzed in six pairwise comparisons, R2-21_6h vs. Con_6h comparison, R2-21_6d vs. Con_6d, R2-21_15d vs. Con_15d, XHYN2_6h vs. Con_6h, XHYN2_6d vs. Con_6d, XHYN2_15d vs. Con_15d. R2-21_6h, R2-21_6d, and R2-21_15d represent leaf samples from Epi R2-21 treatments collected at 6 h, 6 d, and 15 d after inoculation, respectively; XHYN2_6h, XHYN2_6d, and XHYN2_15d represent leaf samples from Alt XHYN2 treatments collected at 6 h, 6 d, and 15 d after inoculation, respectively; and Con_6h, Con_6d, and Con_15d represent leaf samples from the control collected at 6 h, 6 d, and 15 d after inoculation, respectively.

**Figure 2 jof-09-01154-f002:**
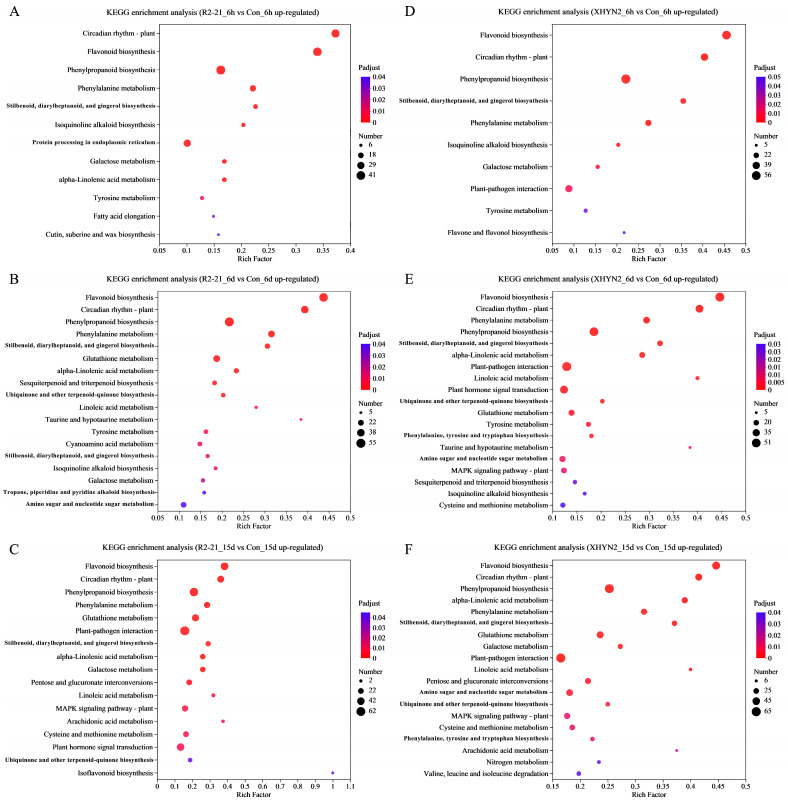
KEGG pathway enrichment analysis of up-regulated DEGs of the tissue cultured seedlings after 6 h, 6 d, and 15 d inoculation with Epi R2-21 and Alt XHYN2. DEGs were analyzed in six pairwise comparisons, R2-21_6h vs. Con_6h comparison, R2-21_6d vs. Con_6d, R2-21_15d vs. Con_15d, XHYN2_6h vs. Con_6h, XHYN2_6d vs. Con_6d, XHYN2_15d vs. Con_15d. R2-21_6h, R2-21_6d, and R2-21_15d represent leaf samples from Epi R2-21 treatments collected at 6 h, 6 d, and 15 d after inoculation, respectively; XHYN2_6h, XHYN2_6d, and XHYN2_15d represent leaf samples from Alt XHYN2 treatments collected at 6 h, 6 d, and 15 d after inoculation, respectively; and Con_6h, Con_6d, and Con_15d represent leaf samples from the control collected at 6 h, 6 d, and 15 d after inoculation, respectively. (**A**), (**B**) and (**C**) represents the KEGG enrichment analysis of up-regulated DEGs of the tissue cultured seedlings after 6 h, 6 d, and 15 d inoculation with Epi R2-21, respectively. (**D**), (**E**) and (**F**) represents the KEGG enrichment analysis of up-regulated DEGs of the tissue cultured seedlings after 6 h, 6 d, and 15 d inoculation with Alt XHYN2, respectively.

**Figure 3 jof-09-01154-f003:**
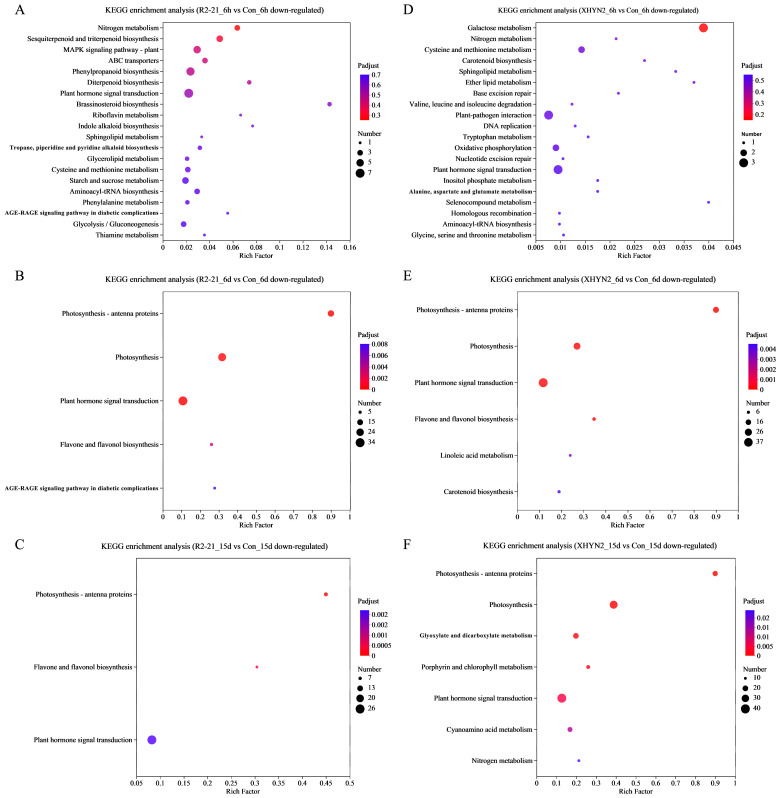
KEGG pathway enrichment analysis of down-regulated DEGs of the tissue cultured seedlings after 6 h, 6 d, and 15 d inoculation with Epi R2-21 and Alt XHYN2. DEGs were analyzed in six pairwise comparisons, R2-21_6h vs. Con_6h comparison, R2-21_6d vs. Con_6d, R2-21_15d vs. Con15d, XHYN2_6h vs. Con_6h, XHYN2_6d vs. Con_6d, XHYN2_15d vs. Con_15d. R2-21_6h, R2-21_6d, and R2-21_15d represent leaf samples from Epi R2-21 treatments collected at 6 h, 6 d, and 15 d after inoculation, respectively; XHYN2_6h, XHYN2_6d, and XHYN2_15d represent leaf samples from Alt XHYN2 treatments collected at 6 h, 6 d, and 15 d after inoculation, respectively; and Con_6h, Con_6d, and Con_15d represent leaf samples from the control were collected at 6 h, 6 d, and 15 d after inoculation, respectively. (**A**), (**B**) and (**C**) represents the KEGG enrichment analysis of down-regulated DEGs of the tissue cultured seedlings after 6 h, 6 d, and 15 d inoculation with Epi R2-21, respectively. (**D**), (**E**) and (**F**) represents the KEGG enrichment analysis of down-regulated DEGs of the tissue cultured seedlings after 6 h, 6 d, and 15 d inoculation with Alt XHYN2, respectively.

**Figure 4 jof-09-01154-f004:**
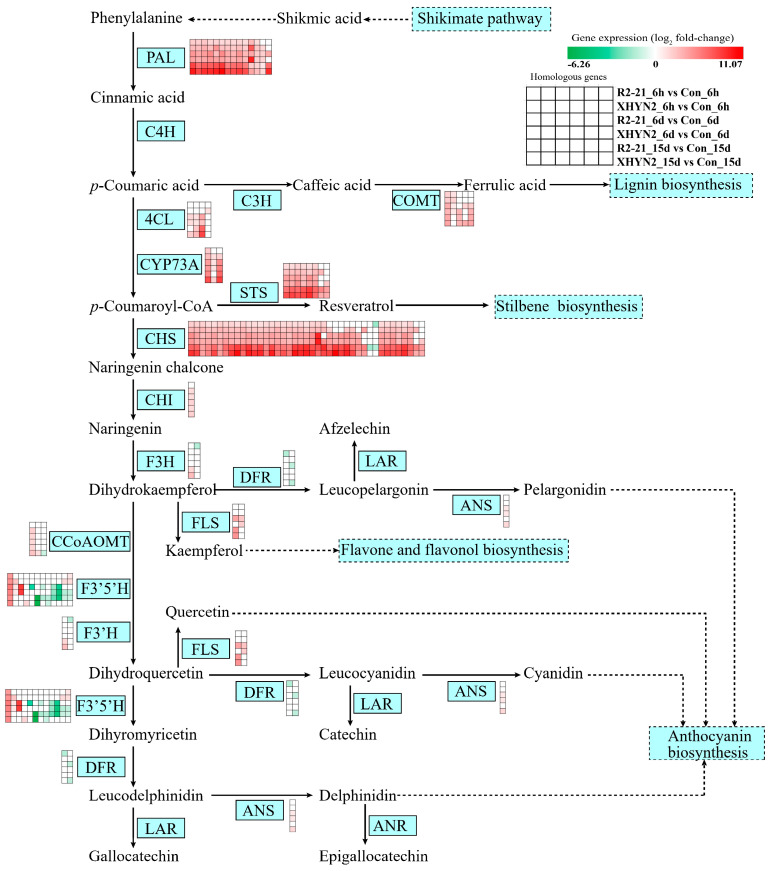
DEGs involved in secondary metabolism in tissue cultured seedlings after 6 h, 6 d, and 15 d infection with Epi R2-21 or Alt XHYN2. Dashed lines represent omitted steps. Heatmap colors represent the gene expression (log_2_ fold change) for each comparison (Appendix A). DEGs were analyzed in six pairwise comparisons, R2-21_6h vs. Con_6h comparison, R2-21_6d vs. Con_6d, R2-21_15d vs. Con_15d, XHYN2_6h vs. Con_6h, XHYN2_6d vs. Con_6d, XHYN2_15d vs. Con_15d. R2-21_6h, R2-21_6d, and R2-21_15d represent leaf samples from Epi R2-21 treatments collected at 6 h, 6 d, and 15 d after inoculation, respectively; XHYN2_6h, XHYN2_6d, and XHYN2_15d represent leaf samples from Alt XHYN2 treatments collected at 6 h, 6 d, and 15 d after inoculation, respectively; and Con_6h, Con_6d, and Con_15d represent leaf samples from the control collected at 6 h, 6 d, and 15 d after inoculation, respectively.

**Figure 5 jof-09-01154-f005:**
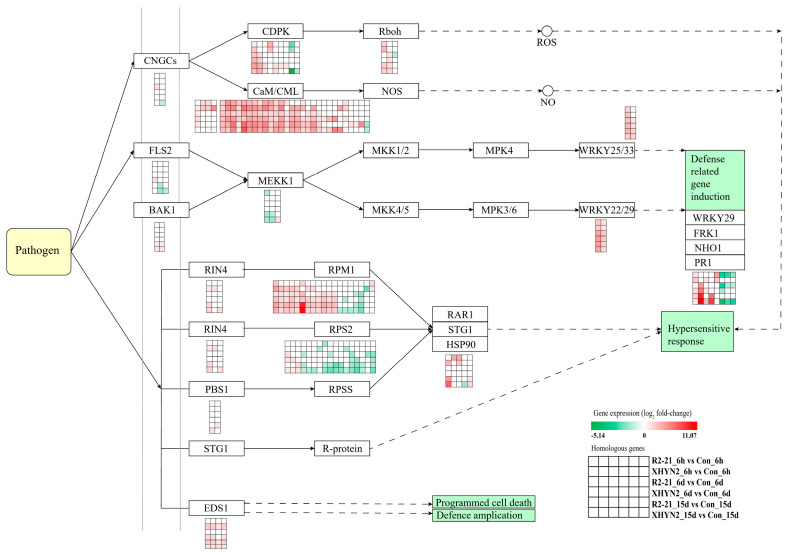
DEGs involved in the plant–pathogen interaction pathway in tissue cultured seedlings after 6 h, 6 d, and 15 d infection with Epi R2-21 or Alt XHYN2. DEGs were analyzed in six pairwise comparisons, R2-21_6h vs. Con_6h comparison, R2-21_6d vs. Con_6d, R2-21_15d vs. Con_15d, XHYN2_6h vs. Con_6h, XHYN2_6d vs. Con_6d, XHYN2_15d vs. Con_15d. R2-21_6h, R2-21_6d, and R2-21_15d represent leaf samples from Epi R2-21 treatments collected at 6 h, 6 d, and 15 d after inoculation, respectively; XHYN2_6h, XHYN2_6d, and XHYN2_15d represent leaf samples from Alt XHYN2 treatments collected at 6 h, 6 d, and 15 d after inoculation, respectively; and Con_6h, Con_6d, and Con_15d represent leaf samples from the control collected at 6 h, 6 d, and 15 d after inoculation, respectively.

**Figure 6 jof-09-01154-f006:**
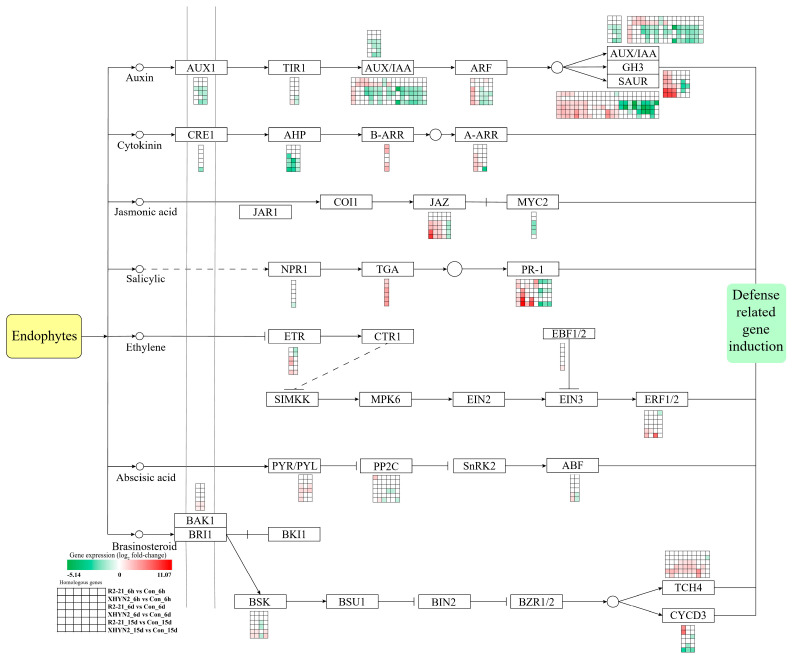
DEGs involved in the plant hormone signal transduction pathway in tissue cultured seedlings after 6 h, 6 d, and 15 d infection with Epi R2-21 or Alt XHYN2. DEGs were analyzed in six pairwise comparisons, R2-21_6h vs. Con_6h comparison, R2-21_6d vs. Con_6d, R2-21_15d vs. Con_15d, XHYN2_6h vs. Con_6h, XHYN2_6d vs. Con_6d, XHYN2_15d vs. Con_15d. R2-21_6h, R2-21_6d, and R2-21_15d represent leaf samples from Epi R2-21 treatments collected at 6 h, 6 d, and 15 d after inoculation, respectively; XHYN2_6h, XHYN2_6d, and XHYN2_15d represent leaf samples from Alt XHYN2 treatments collected at 6 h, 6 d, and 15 d after inoculation, respectively; and Con_6h, Con_6d, and Con_15d represent leaf samples from the control collected at 6 h, 6 d, and 15 d after inoculation, respectively.

**Figure 7 jof-09-01154-f007:**
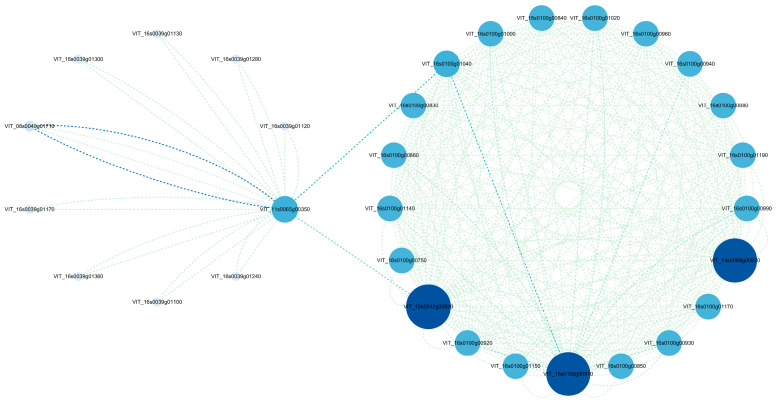
Interaction analysis of DEGs that were associated with secondary metabolism and involved in tissue cultured seedlings defense to Epi R2-21 or Alt XHYN2. Node color from light to dark represents lowest to highest betweenness centrality. The size of each node corresponds to the degree (the number of connections).

**Table 1 jof-09-01154-t001:** The number of DEGs enriched significantly in the KEGG pathway of the tissue cultured seedlings (*p*-adjust < 0.05). DEGs were analyzed in six pairwise comparisons, R2-21_6h vs. Con_6h comparison, R2-21_6d vs. Con_6d, R2-21_15d vs. Con_15d, XHYN2_6h vs. Con_6h, XHYN2_6d vs. Con_6d, XHYN2_15d vs. Con_15d. R2-21_6h, R2-21_6d, and R2-21_15d represent leaf samples from Epi R2-21 treatments collected at 6 h, 6 d, and 15 d after inoculation, respectively; XHYN2_6h, XHYN2_6d, and XHYN2_15d represent leaf samples from Alt XHYN2 treatments collected at 6 h, 6 d, and 15 d after inoculation, respectively; and Con_6h, Con_6d, and Con_15d represent leaf samples from the control collected at 6 h, 6 d, and 15 d after inoculation, respectively. “/” represents no DEG enriched significantly in the KEGG pathway.

KEGG Pathway	The DEGs Number					
	R2-21_6h vs. Con_6h	XHYN2_6h vs. Con_6h	R2-21_6d vs. Con_6d	XHYN2_6d vs. Con_6d	R2-21_15d vs. Con_15d	XHYN2_15d vs. Con_15d
Up-regulated						
Flavonoid biosynthesis	38	51	49	50	43	50
Circadian rhythm-plant	35	38	37	38	34	39
Phenylpropanoid biosynthesis	41	56	55	47	53	64
Phenylalanine metabolism	21	26	30	28	27	30
Stilbenoid, diarylheptanoid, and gingerol biosynthesis	14	22	19	20	18	23
Flavone and flavonol biosynthesis	/	5	/	/	/	/
Linoleic acid metabolism	/	/	7	10	8	10
alpha-Linolenic acid metabolism	13	/	18	22	20	30
Plant–pathogen interaction	/	35	/	51	62	65
Isoquinoline alkaloid biosynthesis	11	11	10	9	2	/
Galactose metabolism	13	12	12	/	20	21
Protein processing in endoplasmic reticulum	29	/	/	/	/	/
Tyrosine metabolism	11	11	14	15	/	/
Fatty acid elongation	7	/	/	/	/	/
Cutin, suberine, and wax biosynthesis	6	/	/	/	/	/
Glutathione metabolism	/	/	31	23	36	39
Sesquiterpenoid and triterpenoid biosynthesis	/	/	15	12	/	/
Ubiquinone and other terpenoid-quinone biosynthesis	/	/	13	13	12	16
Taurine and hypotaurine metabolism	/	/	5	5	/	/
Cyanoamino acid metabolism	/	/	15	/	/	/
Phenylalanine, tyrosine and tryptophan biosynthesis	/	/	12	13	/	16
Tropane, piperidine, and pyridine alkaloid biosynthesis	/	/	10	/	/	/
Amino sugar and nucleotide sugar metabolism	/	/	22	24	/	36
Plant hormone signal transduction	/	/	/	39	42	/
MAPK signaling pathway-plant	/	/	/	21	27	30
Cysteine and methionine metabolism	/	/	/	17	23	26
Pentose and glucuronate interconversions	/	/	/	/	23	27
Arachidonic acid metabolism	/	/	/	/	6	6
Nitrogen metabolism	/	/	/	/	/	11
Valine, leucine and isoleucine degradation	/	/	/	/	/	16
Down-regulated						
Photosynthesis–antenna proteins	/	/	18	18	9	18
Photosynthesis	/	/	27	23	/	33
Plant hormone signal transduction	/	/	34	37	26	40
Flavone and flavonol biosynthesis	/	/	6	8	7	/
AGE-RAGE signaling pathway in diabetic complications	/	/	5	/	/	/
Linoleic acid metabolism	/	/	/	6	/	/
Carotenoid biosynthesis	/	/	/	7	/	/
Glyoxylate and dicarboxylate metabolism	/	/	/	/	/	20
Porphyrin and chlorophyll metabolism	/	/	/	/	/	13
Cyanoamino acid metabolism	/	/	/	/	/	17
Nitrogen metabolism	/	/	/	/	/	10

## Data Availability

The datasets for this study can be found in the Sequence Read Archive (SRA) under the study accession number PRJNA667208.

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
