# Peer review of "The Interactions between Two Fungal Endophytes Epicoccum layuense R2-21 and Alternaria alternata XHYN2 and Grapevines (Vitis vinifera) with De Novo Established Symbionts under Aseptic Conditions"

_jof, 2023, doi:10.3390/jof9121154_

Round 1
Reviewer 1 Report
Comments and Suggestions for Authors
See the attached file about comments and suggestions.

The manuscript needs to be edited by English editor for sound English statements and phrases.
Author Response
Response to Reviewer 1’ Comments
Dear editor and reviewer,
the authors are greatly appreciated for those valuable comments, which were all helpful to improve our manuscript. The present version has been revised according to comments, and the present version has been fixed the the syntax and grammatical errors by a professional editing service, American Journal Experts (AJE). The point to point response to comments was attached as followed:
Title
Comments:
The Interactions between Fungal Endophytes and Grapevines with the de novo Established Symbionts
- Correct words spacing in the title,
- It will be clearer, and more indicative mentioning the names of those two fungi in the title rather than leaving it very broad by saying endophytes in general.
- Proposed title;
The Interactions between two Fungal Endophytes Epicoccum layuense R2-21 and Alternaria alternata XHYN2 and Grapevines (Vitis vinifera) with the de novo Established Symbionts under aseptic conditions.
- Also, the names of those fungi can be useful keywords.
The use description, ‘symbiosis’ is not exactly applicable to the interaction between those fungi and the host. They are only endophytes.
Response: The title of the manuscript has been changed to "The Interactions between two Fungal Endophytes Epicoccum layuense R2-21 and Alternaria alternata XHYN2 and Grapevines (Vitis vinifera) with De Novo Established Symbionts under Aseptic Conditions". And the keywords in the text have been changed accordingly. Thank you very much!
Introduction
Comments:
- Line 65. Citation (7-10) needs to be changed to (7and 10) or (7,9) as it means two references only and not references 7 to 10.
Response: We've checked the citation and we didn't cite the references as (7-10) in line 65 or in the Introduction section.
Comments:
- Starting here and for the rest of the manuscript the scientific names of those two fungi are completely ignored and replaced by their strain’s alphabets and numbers. It is suggested, once the name and its strain’s indication are affixed, then it will be much smoother and closely related to use the generic name of those fungi (Epicoccum, and Alternaria) though out the manuscript.
- For instance starting on line 85 of the Materials and Methods it would be shown as follows; Endophytic fungi Epicoccum layuense, isolate R2-21 (Epi R2-21)and Alternaria alternata isolate XHYN2 (Alt XNH2).
Response: Thank you for the sensible suggestion, we have changed the fungi names to Epicoccum layuense R2-21 (Epi R2-21) and Alternaria alternata XHYN2 (Alt XHYN2) throughout the text.
Comments:
- And of course, those isolates should be indicated how were they identified to their species level for this work.
Response: Thanks for your pointing out, the section has been added in the present version.
Materials and Methods
Comments:
- Note the point raised above about the fungal names used in this work.
- Figure S1. Here also the names of the two fungi are better to be used rather than the letters and numbers.
Response: Thank you for this good suggestion, we have changed the names of those two fungi throughout the text.
Comments:
- Line 64. “Aseptic grapevine seedlings were cultured in a light culture room at 26 °C with 12/12 light/dark cycles for 2 months.” Production of axenic, germ-free, healthy plant seedlings is a very complicated and high axenic technology. A relevant reference needs to be cited here. And good presentation about how those germ-free seedlings were produced.
Response: Sorry for our not updated term “aseptic seedling”. Traditionally, we called the successfully established in vitro cultured plantlets under aseptic condition as “aseptic seedlings”. With the development of cultivation-independent technology, we have accepted the fact that in vitro cultured plant tissues or plantlets, even these cultures have undergone decades of subculture under aseptic conditions, were also the host of diverse endophytic microorganisms (Thomas and Franco, 2021. Xiang et al., 2023). Therefore, the aseptic grapevine seedlings used in present study are just the in vitro grapevine plantlets without the contamination of any culturable microbes, not the real axenic, germ-free seedlings. In this version, we have changed the term as in vitro cultured vine plantlet or tissue culture seedlings. In addition, how to establish these tissue culture plantlets was included in the manuscript.
Thomas P, Franco C, 2021. Intracellular Bacteria in Plants: Elucidation of Abundant and Diverse Cytoplasmic Bacteria in Healthy Plant Cells Using In Vitro Cell and Callus Cultures. Microorganisms 269, 1-27.
Xiang SY, Wang YT, Chen CX, et al., 2023. Dominated “Inheritance” of Endophytes in Grapevines from Stock Plants via In Vitro-Cultured Plantlets: The Dawn of Plant Endophytic Modifications. Horticulturae 9, 180.
Comments:
- The pictures of those fungi growing on agar are not indicative enough for the fungi. It is needed to be supported by pictures of their specific reproductive structures.
Response: Microscopic study to observe the reproductive structures of Epicoccum layuense R2-21 and Alternaria alternata XHYN2 has been added as Figure S1 in this version.
Comments:
- Note the agar media in the control jar is clear while the one in the treatment jars is off-color. This needs to be clarified further. This needs to be clarified and explained why?
Response: In the text, we have added the following section in the results section: However, the MS media inoculated with Epicoccum layuense R2-21 or Alternaria alternata XHYN2 were slightly off-coloured, with a little brown (Figure S1). There may be two reasons for this: firstly, the suspensions of Epicoccum layuense R2-21 and Alternaria alternata XHYN2 may have fallen onto the medium, causing the growth of fungal mycelia on the plate; secondly, inoculation with Epicoccum layuense R2-21 or Alternaria alternata XHYN2 may have caused the secretion of some colored secondary metabolites.
Comments:
- Line 90. Were those fungi sporulating during that growth time range of 3-5 days. The inoculum used may have been not only “fungal mycelium”?
- Saying; (final concentration was 2.5 g/L), did prepare 1L. Or how was it reached to that value?
- “Aseptic seedlings were inoculation with mycelium suspensions of Epicoccum R2-21 and Alternaria XHYN2 respectively to establish endophytes-host symbionts.” Hence it may be “fungal growth mass” rather than mycelium only.
- It is not clear how the “inoculation” performed. It is important to explain how the inoculation was done.
Response: (1) Thank you for pointing out! We have changed “fungal mycelium” to “fungal growth mass” in the text. (2) We weighed 0.125 g of fungal growth mass and suspended in 50 ml of 0.9% normal saline, giving a final concentration of 2.5 g/L. (3) In the text, “fungal mycelium” has changed to “fungal growth mass”. (4) In our study, tissue seedlings were smear inoculated with a sterilized cotton swab and added in the text.
Comments:
- Line 95. “All samples of aseptic seedlings were harvested and washed off leaf surface fungi with sterile water”. Did you mean, leaves samples from treated and control were collected. Leaf surfaces were thoroughly cleaned off the residual fungal material….
- Cleaning the leaf surfaces for the purpose of the present investigating the endophyte should be done very thoroughly to the extent it may be surface sterilized. That is leaving no residual fungal material behind.
- Furthermore, it is postulated a new endophytic relationship was constructed. However, that should have been documented by stringent reisolating from completely surface sterilized plant materials, leaves. How close the following was observed as described in the reference number
15. (Surface sterilization and isolation of endophytic fungi Five grams of fresh root material were washed carefully under running tap water. The roots were macroscopically inspected. Areas with lesions and discolorations were discarded so that only healthy root material was used. The sterilization procedure contained a 1 min step in ethanol (70%), 4.5 min sodium hypochlorite (5% active chlorine and three washing steps for at least 5 min with sterile water).
- Also, leave samples were collected after 15 days of inoculation and supposedly tested and documented for the reconstruction of endophytic condition.
- Now, were that also tested for that condition during and after the two months of incubation?
Response: In this study, all samples of grapevine seedlings were harvested at different days after inoculation, and the leaf surface fungi were washed 3 times with sterile water. Leaves washed 3 times only used for RNA sequencing assay, leaves fro reisolating the inoculated fungi were further done surface sterilizations.
Each leaf was cut into two parts along the main vein. One part of the leaf was used for RNA sequencing, and the other part was used to determine the isolation rates of fungal endophytes using the tissue patch method as described by Götz et al (2006). Surface sterilization of the leaves was performed using 75% ethanol for 30 s, 3% sodium hypochlorite for 20 min, and three washes with sterilized water. After surface sterilization, leaves were cut into 0.5 cm segments with sterile scissors and placed on the isolation medium (potato dextrose agar medium). Prior to initial plating, several samples were imprinted onto media and these imprinted plates were monitored for lack of fungal growth to ensure the effectiveness of the surface sterilization. Symbiosis rates were calculated as the percentage of emerged fungal colonies per leaf patch and were used to describe the efficiency of fungal endophytes symbiosis. Fungal colonies were identified using ITS DNA sequences.
To reduce the length of the article, the reference has been inserted in the present version and gave a brief description.
Comments:
- It is not clear on lines 84-85 why and what for plants were kept for two months when all sampling was made within 15 days.
Response: Tissue culture plantlets sub-cultured for 2 months with 7 to 10 expanded leaves were prepared for the experiments, and after inoculation with Epicoccum layuense R2-21 and Alternaria alternata XHYN2 for 15 days, all samples of seedlings were harvested for analyses.
Comments:
- There is a missing treatment of inoculating the axenic vine grape seedlings with an inoculum made of both fungi at the same time.
- Note, there is a missing control treatment. There should be a treatment using not alive fungal growth mass. This would have ruled out any effect of possible extracellular metabolites introduced with the inoculum.
Response: In this study, we carried out the experiments in which grapevine seedlings were inoculated with Epicoccum layuense R2-21 and Alternaria alternata XHYN2 respectively at three different time points (6 h, 6 d, 15 d). Indeed, we sincerely admit that the treatment of inoculating the grapevine seedlings with an inoculum composed of both fungi at the same time or the treatment of not-living fungal growth mass is more comprehensive or rigorous. Nevertheless, please understand that it is very difficult to modify these data at the present time. The authors thank the reviewer for the good suggestions.
Comments:
- Note, ‘aseptic seedlings’ (germ-free or axenic) were produced and then according to this work became “untreated”, ‘control’ seedlings and treated seedlings (treated with the fungi). The term: sceptic seedlings is used in a confusing manner. So, it is needed to correct that throughout the manuscript.
Response: Sorry for our not being rigorous enough, we have revised the text accordingly, unified with this name as “tissue grapevine seedlings”. Thank you!
Comments:
- Lines 98-99. If that molecular analysis was performed why not fungi identified to their species level?
Response: The two fungi were identified to species level as Epicoccum layuense and Alternaria alternata.
Comments:
- Lines 99-103, and for the rest of the manuscript. It would be clearer and easier to the readers to use the fungal name(s) or its abbreviation in conjunction with time of incubation and/or any treatment. Here, the fungal real names were ignored and instead too many numbers and letter introduced.
- “These samples were defined as R2-21_6h, R2-21_6d, R2-21_15d (strain R2-21 group, aseptic seedlings after co-culture with R2-21 for 6 h, 6 d, 15 d, respectively); XHYN2_6h, XHYN2_6d, XHYN2_15d (strain XHYN2 group, aseptic seedlings after co-culture (confusing!!) with XHYN2 for 6 h, 6 d, 15 d, respectively); and Con_6h, 102 Con_6d, Con_15d (control group).”
How about saying; Leves samples from Epicoccum (or Epi) and Alternaria (or Alt) treatments and the control were collected at 6h, 6d and 15d after innovation.
Response: We have revised the text accordingly as follows: These samples were defined as R2-21_6h, R2-21_6d, and R2-21_15d (leaf samples from Epicoccum layuense R2-21 treatments collected at 6 h, 6 d and 15 d after inoculation); XHYN2_6h, XHYN2_6d, and XHYN2_15d (leaf samples from Alternaria alternata XHYN2 treatments collected at 6 h, 6 d and 15 d after inoculation); and Con_6h, Con_6d, and Con_15d (leaf samples from the control collected at 6 h, 6 d and 15 d after inoculation). Thank you very much!
Comments:
- Line 124 and throughout the manuscript the acronym name needs to be preceded by its full names. Like Identification of Differentially Expressed Genes (DEGs), as an example.
Response: We have added the full names of abbreviations in the text.
Results
Comments:
- Lines 148-149. Figure 1. PCA (A), DEGs analysis (B), and Venn analysis (C, D) of the aseptic grapevine seedlings after 6 h, 6 d, and 15 d inoculation with or without fungal endophytes (this is not sound phrasing) R2-21 or XHYN2. It is either inoculated or not!!
- The above caption needs to be rewritten according to points raised above about abbreviations.
The figure’s caption and table’s title should be descriptive enough to make that figure and table can stand alone and well comprehended by the reader.
Response: The caption of the figure and the title of the table have been rewritten accordingly. Thank you very much!
Comments:
- Lines 189-194. Not clear, needs to be rewritten, rephrased.
Response: We have revised accordingly. Thank you!
Comments:
- Table 1. The table title needs to be reconsidered a according to the note raised above.
- Also, clarify what is meant by (vs) and how was it derived or figured out.
Response: The title of Table 1 have been rewritten accordingly. Thank you!
Discussion
Comments:
There is a need for a conclusion statements pulling-out the main lines of such interaction.
Response: Thanks for the reminder, and we have added a conclusion to the text.

Reviewer 2 Report
Comments and Suggestions for Authors
Dear authors, the manuscript "The Interactions between Fungal Endophytes and Grapevines with the de novo Established Symbionts" presents a good research in the field of grapevines and their reaction to microbial symbionts.
there are some suggestions that can improve the work of the authors and increase the quality of the manuscript.
At the end of the Introduction, expand the last lines to clearly state, in separate sentences, the aim and the objectives of your work. You can use the flow of the Results section to establish the objectives/hypotheses.
Increase the quality of the graphs to make them more clear. They are interesting and should be presented in a higher resolution.
Add a conclusion section where to point each of your main findings, preferably with some values.
Overall, the manuscript is well written and each section have an appropriate length.
Author Response
Response to Reviewer 2’ Comments
Dear editor and reviewer,
the authors are greatly appreciated for those valuable comments, which were all helpful to improve our manuscript. The present version has been revised according to comments, and the present version has been fixed the the syntax and grammatical errors by a professional editing service, American Journal Experts (AJE). The point to point response to comments was attached as followed:
Dear authors, the manuscript "The Interactions between Fungal Endophytes and Grapevines with the de novo Established Symbionts" presents a good research in the field of grapevines and their reaction to microbial symbionts.
there are some suggestions that can improve the work of the authors and increase the quality of the manuscript.
Comments:
At the end of the Introduction, expand the last lines to clearly state, in separate sentences, the aim and the objectives of your work. You can use the flow of the Results section to establish the objectives/hypotheses.
Response: Thank you for your good suggestion, and we have expanded the last lines at the end of the introduction.
Comments:
Increase the quality of the graphs to make them more clear. They are interesting and should be presented in a higher resolution.
Response: Thank you for your suggestion. We have remade the figures in the text.
Comments:
Add a conclusion section where to point each of your main findings, preferably with some values.
Response: Thank you for your suggestion, and we have added a conclusion in the text.
Comments:
Overall, the manuscript is well written and each section have an appropriate length.
Response:
Thank you for your positive feedback on our manuscript. In order to bring the text closer to the scientific style and rules of the English language, the present version has been edited by a professional editing service, American Journal Experts (AJE), to correct the syntax and grammatical errors.
Reviewer 3 Report
Comments and Suggestions for Authors
The manuscript with the title “The Interactions between Fungal Endophytes and Grapevines with the de novo Established Symbionts”, presents the results of 2-month old Vitis vinifera L. cv. Cabernet Sauvignon plantlets inoculated with Epicoccum and Alternaria cultured for 6 h, 6 d, 15 d, with main focus on molecular responses at transcriptomic levels due to fungal colonization. A total of 18 5748 and 5817 differentially expressed genes (DEGs) were separately initiated by fungi application. Findings suggest the plants can have enhanced synthesis of some compounds due to presence of such fungi, due to activation of stress-associated plant secondary metabolism.
Introduction – some information from Lines 76-81 should be at Material and Method section.
Figures 2 and 3 components are too small to read.
Best regards.
Comments on the Quality of English Languagesmall grammar improvements
Author Response
Response to Reviewer 3’ Comments
Dear editor and reviewer,
the authors are greatly appreciated for those valuable comments, which were all helpful to improve our manuscript. The present version has been revised according to comments, and the present version has been fixed the the syntax and grammatical errors by a professional editing service, American Journal Experts (AJE). The point to point response to comments was attached as followed:
The manuscript with the title “The Interactions between Fungal Endophytes and Grapevines with the de novo Established Symbionts”, presents the results of 2-month old Vitis vinifera L. cv. Cabernet Sauvignon plantlets inoculated with Epicoccum and Alternaria cultured for 6 h, 6 d, 15 d, with main focus on molecular responses at transcriptomic levels due to fungal colonization. A total of 18 5748 and 5817 differentially expressed genes (DEGs) were separately initiated by fungi application. Findings suggest the plants can have enhanced synthesis of some compounds due to presence of such fungi, due to activation of stress-associated plant secondary metabolism.
Comments:
Introduction – some information from Lines 76-81 should be at Material and Method section.
Response: In the text, we have added the relevant information in the “Materials and Methods” section.
Comments:
Figures 2 and 3 components are too small to read.
Response: Thank you for your suggestion. We have remade the figures in the text.
Besides, in order to bring the text closer to the scientific style and rules of the English language, the present version has been edited by a professional editing service, American Journal Experts (AJE), to correct the syntax and grammatical errors.

Reviewer 4 Report
Comments and Suggestions for Authors
The manuscript titled ¨ The Interactions between Fungal Endophytes and Grapevines with the de novo Established Symbionts¨ by Pan and coauthors investigates the transcriptomic changes of grapevine seedlings to the inoculation of two endophytic fungi strains R2-21 (Epicoccum layuense) and XHYN2 (Alternaria alternata) at three different time-points.
The work in general sounds good, analysis is described in good detail and graphics and statistical analysis are of great quality, repetitions number is okay.
Results are interesting from the point of view that fungal endophytes induce a modification in the secondary metabolism.
The quality of the RNAseq was also validated by qPCR; however, I would encourage the authors to present those results in the body of the text with a nice graphic.
Writing need very few improvements, see below.
But it would be nice to see the grapevine seedlings pictures, I do not know small or big they are at the day of analysis. Perhaps in supplementary information.
Minor comments:
Please make bigger the Figures, it is hard to read the axis titles. You still have half of the page in blank.
Please correct spaces between the title words
Please check out initial names in affiliations
L91.92: Aseptic seedlings were inoculation with mycelium suspensions of R2-21 and XHYN2 respectively… PLEASE CORRECT inoculation to ¨inoculated¨
L322: Please rewrite to give fluidity of the sentence: ¨ …with a positive effect on increasing cryptotanshinone production [17]. And endophytic fungi isolated from grapevines demonstrated intriguing resveratrol-producing ability…¨
Author Response
Response to Reviewer 4’ Comments
Dear editor and reviewer,
the authors are greatly appreciated for those valuable comments, which were all helpful to improve our manuscript. The present version has been revised according to comments, and the present version has been fixed the the syntax and grammatical errors by a professional editing service, American Journal Experts (AJE). The point to point response to comments was attached as followed:
The manuscript titled ¨ The Interactions between Fungal Endophytes and Grapevines with the de novo Established Symbionts¨ by Pan and coauthors investigates the transcriptomic changes of grapevine seedlings to the inoculation of two endophytic fungi strains R2-21 (Epicoccum layuense) and XHYN2 (Alternaria alternata) at three different time-points.
Comments:
The work in general sounds good, analysis is described in good detail and graphics and statistical analysis are of great quality, repetitions number is okay.
Results are interesting from the point of view that fungal endophytes induce a modification in the secondary metabolism.
Response:
Thank you for the positive evaluation to our manuscript. In order to bring the text closer to the scientific style and rules of the English language, the present version has been edited by a professional editing service, American Journal Experts (AJE), to correct the syntax and grammatical errors.
Comments:
The quality of the RNAseq was also validated by qPCR; however, I would encourage the authors to present those results in the body of the text with a nice graphic.
Response: To reduce the length of the article, the figure of qRT-PCR is shown in Figure S2. Thank you!
Comments:
Writing need very few improvements, see below.
But it would be nice to see the grapevine seedlings pictures, I do not know small or big they are at the day of analysis. Perhaps in supplementary information.
Response: To reduce the length of the article, the pictures of the grapevine seedlings inoculated with Epicoccum layuense R2-21 and Alternaria alternata XHYN2 after 15 d are presented as Figure S1 in the supplementary material.
Minor comments:
Please make bigger the Figures, it is hard to read the axis titles. You still have half of the page in blank.
Response: Thank you for your suggestion. We have remade the figures in the text.
Please correct spaces between the title words
Response: Thanks for pointing this out! We have fixed it in the current version.
Please check out initial names in affiliations
Response: We have checked and revised in the present version.
L91.92: Aseptic seedlings were inoculation with mycelium suspensions of R2-21 and XHYN2 respectively… PLEASE CORRECT inoculation to ¨inoculated¨
Response: Sorry for our mistake, we have revised in the text.
L322: Please rewrite to give fluidity of the sentence: ¨ …with a positive effect on increasing cryptotanshinone production [17]. And endophytic fungi isolated from grapevines demonstrated intriguing resveratrol-producing ability…¨
Response: This sentence has been rewritten accordingly. Thank you!

Round 2
Reviewer 1 Report
Comments and Suggestions for Authors
It is significant to think of including those controls in future work with the endophytes.